# Changes in anemia prevalence and the proportion of anemia associated with iron deficiency or inflammation in young children residing in Puno, Peru: Analysis using new World Health Organization guidelines for defining anemia

Cinthya Vásquez-Velásquez[1]*, Benita Maritza Choque-Quispe[2], Parminder S. Suchdev[3,4], Chris A. Rees[5,6], Vilma Tapia[7], Yi-An Ko[8], Gustavo F. Gonzales[1]

**1** Carrera de Medicina Humana, Facultad de Ciencias de la Salud, Universidad San Ignacio de Loyola, Lima, Peru, **2** Facultad de Ciencias de la Salud, Universidad Nacional del Altiplano, Puno, Peru, **3** Hubert Department of Global Health, Emory University, Rollins School of Public Health, Atlanta, Georgia, United States of America, **4** Department of Pediatrics, Emory University, Atlanta, Georgia, United States of America, **5** Division of Pediatric Emergency Medicine, Emory University School of Medicine, Atlanta, Georgia, United States of America, **6** Department of Emergency Medicine, Children's Healthcare of Atlanta, Atlanta, Georgia, United States of America, **7** Laboratorio de Endocrinología y Reproducción, Departamento de Ciencias Biológicas y Fisiológicas, Facultad de Ciencias e Ingeniería, Universidad Peruana Cayetano Heredia, Lima, Peru, **8** Department of Biostatistics and Bioinformatics, Rollins School of Public Health, Emory University, Atlanta, Georgia, United States of America

* cvasquezv@usil.edu.pe

## Abstract

### Background

In 2024, the World Health Organization (WHO) introduced new hemoglobin cutoffs for diagnosing anemia. The WHO also incorporated revised altitude adjustments and lowered thresholds to diagnose anemia for children aged 6–23 months. Puno, Peru has historically reported the highest prevalence of anemia in the country, exceeding 70% in infants and young children.

### Objective

To assess the impact of the new WHO cutoffs on anemia prevalence and evaluate whether they affected the proportion of anemia attributable to iron deficiency (ID), inflammation, and other causes.

### Methods

A cross-sectional study was conducted among 310 children aged 6–59 months in Puno, Peru. Participants were recruited via convenience sampling during routine medical check-ups. Venous blood samples were analyzed using an automated hemoglobin analyzer and serum biomarker evaluations. Anemia prevalence was

**Data availability statement:** Data described in the paper has been made publicly and freely available without restriction at DOI 10.5281/zenodo.14183957., URL: https://zenodo.org/records/14183958.

**Funding:** This study was supported by the Vice-Rectorate of Research of the Universidad Nacional del Altiplano, Puno, Peru in the form of a grant awarded to BMCh-Q. Also, The National Institutes of Health in the form of a grant awarded to CAR (K23HL173694). The funders had no role in study design, data collection and analysis, decision to publish, or preparation of the manuscript.

**Competing interests:** The authors declare no conflicts of interest.

determined based on WHO guidelines for children aged 6−59 months (and 6−23 months and 24−59 months as subgroups). The ratio of anemia due to ID (Ferritin <12 ng/mL) or inflammation (IL-6 > 60 pg/mL) was estimated using adjusted Poisson regression models, reporting prevalence ratios (PR).

## Results

Applying the new WHO guidelines, anemia prevalence changed from 50% to 42.2% in children aged 6−59 months (62% to 47% in children aged 6−23 months and from 45.9% to 40.6% in children aged 24−59 months). The proportion of anemia due to ID was 27.5%, due to inflammation was 45.9%, and due to other causes was 26.6%. ID was significantly associated with anemia in both unadjusted and adjusted analyses (PR: 1.4, 95% CI: 1.1–1.8; PR: 1.32, 95% CI: 1.0–1.7). The 2024 WHO guidelines did not substantially alter the estimated proportion of anemia associated with ID or inflammation.

## Conclusions

Application of the new WHO cutoffs resulted in a lower estimated prevalence of anemia among young children. ID accounted for only a small proportion of cases of anemia, emphasizing the need for further research into other causes of childhood anemia in Peru.

## Introduction

In March 2024, the World Health Organization (WHO) revised the altitude adjustments for hemoglobin (Hb) and updated the age-specific Hb cutoffs for diagnosing anemia in children, which were originally established in 1967 [1]. These new guidelines lowered the Hb cutoffs for diagnosing anemia in young children aged 6–23 months to 10.5 g/dL and maintained the threshold at 11.0 g/dL for children aged 24–59 months [2,3].

These changes were driven by evidence showing that the previous WHO criteria did not accurately reflect age-related changes in Hb ontogeny in young children, whose Hb levels are often lower than 11 g/dL during earlier ages. This discrepancy may partially explain the modest impact of iron interventions during the last 20 years in many countries [4]. Additionally, the updated WHO guidelines emphasize the need to consider causes of anemia beyond iron deficiency (ID), including inflammation, infections, genetic causes, and other nutritional deficiencies. The guidelines also introduced a revised Hb correction factor for altitude of residence, which updates the 1989 adjustments [5]. Specifically, the altitude adjustment has been raised for locations above 500 meters and below 3200 meters, while the adjustment for locations at or above 3200 meters has been reduced. Given these updates, it is necessary to determine their impact on the prevalence of anemia, as well as whether these changes impact the proportion of anemia associated with ID, inflammation, and other etiologies [6].

Peru has one of the highest prevalences of childhood anemia throughout the Americas with historical estimates of anemia among young children aged 6–35 months as high as >70% in its highland regions and >40% across the country [7]. Due to the high prevalence of childhood anemia in Peru, universal iron supplementation has been implemented by the government of Peru for infants under 5 years of age and pregnant women from 14 weeks of gestation [8–12]. This strategy has been implemented based on the assumption that most childhood anemia is due to ID, without high quality studies to determine the cause of anemia among young children in Peru and in other settings globally. Moreover, other studies have shown that infants without anemia who are supplemented with iron may be at risk of adverse effects rather than benefit [13,14].

Our objectives were to 1) evaluate the prevalence of anemia among infants and children residing in the high-altitude region of Puno, Peru using the new WHO Hb cutoffs and altitude adjustments [2] and 2) to determine if the proportion of anemia associated with ID, inflammation, and other causes were reclassified with the application of the new WHO guidelines [6].

## Methods

### Study design

We conducted a cross-sectional study in the 13 provinces of the Puno Region in Peru. Recruitment of infants and children occurred during the first half of 2019.

### Study setting

We conducted our study in Puno, Peru since this region has one of the highest prevalences of anemia in the world (i.e., estimated 70% in children). Puno is in the Southern part of Peru on the border with Bolivia. Puno lies at 12,500 feet (3,810 meters) above sea level in the Andes Mountains. Each region of Puno has several districts located at altitudes ranging from 610 to 4,660 meters above sea level [15] with a population of children aged 6–59 months of 40,162, which is approximately 1.69% of all children in this age group in Peru [16]. The main ethnic group in Puno is a relatively homogeneous ethnic group called Aymara [17]. The Aymara population has long resided in the Peruvian highland and may have a genetic adaptation to high altitude based on prior work [18].

### Study procedures

Convenience sampling was applied for enrollment of children aged 6–59 months who were born in the same place of the study. Young children vaccinated on the day of data collection, those with acute illness (regardless of the need for medication as indicated by clinicians), including children with chronic diseases at the time of the visit to the medical center were not eligible.

Children were recruited by trained personnel in health centers, which they attend for routine checkups after birth as part of the "Growth and Development Control of the Healthy Child" (CRED) program. Routine CRED visits serve to protect children from diseases, detect any health risks early, as well as provide parents or caregivers with breastfeeding counseling, complementary feeding and other parenting issues [19]. CRED is a representative program in Peru, since it is mandatory and serves as the platform for routine vaccinations among children in Peru. During these visits, in addition to routine care, venous blood samples were drawn for Hb measurement and serum biomarkers for this study.

### Sample size

Our sample size was determined from a convenience sample of 310 children that enrolled in our study. *Post hoc*, we determined that, at a significance level of 0.05 (two-sided) and a power of 80%, the minimum required sample size was 122 children per group (anemic and non-anemic) to detect a difference in the prevalence of anemia by the old (2001)

 

WHO guidelines and the new (2024) WHO guidelines. The final recruited sample consisted of 310 children (179 non-anemic children and 131 anemic children).

## Consent and ethical approvals

Parents signed a consent form at the time of enrollment. All study procedures were approved by the Universidad Nacional del Altiplano Review Board (3680–2017-R-UNA) and the Puno Health Direction Center. The Universidad Nacional del Altiplano shared the database with Universidad Peruana Cayetano Heredia (date: 02/09/2019). Approval of the analysis of data was performed by the Institutional Review Board at the Universidad Peruana Cayetano Heredia (Code N°464-20-19). The study was conducted in accordance with the Declaration of Helsinki.

## Biological samples

In each child, 5 mL of venous blood was collected in two tubes, one with anticoagulant for the measurement of Hb and the second without anticoagulant to obtain serum to measure biochemical markers. Venous blood samples were centrifuged at the health center, and the obtained sera were frozen at –40°C and shipped to the city of Lima. All samples were kept frozen until the day of analysis. The time elapsed between the sample collection and the shipment from Lima was < 48 hours. In Lima, samples were analyzed by trained personnel. In each child, Hb was measured in whole blood and biochemical markers of iron status and inflammation were measured in serum.

Whole blood Hb levels were measured using an automated analyzer, CELL-DYN Ruby®. Serum ferritin (SF, ng/mL), soluble transferrin receptor (sTfR, mg/L), interleukin 6 (IL-6, pg/mL), hepcidin (ng/mL) and erythropoietin (EPO, mU/mL) levels were measured with ELISA kits (DRG International, INC, USA).

To define anemia, we used the old and new WHO guidelines for age-specific cutoffs and altitude adjustment (Table 1). For the evaluation of factors associated with anemia, ID was defined as SF concentration <12 ng/mL [20].

We used IL-6 to assess inflammation [16,21]. Because previous studies have applied different IL-6 thresholds (including >50 pg/mL, > 70 pg/mL, and higher values depending on clinical context), and no standardized cutoff exists for defining inflammation-mediated anemia in young children, we evaluated several IL-6 levels within our dataset. Receiver operating characteristic (ROC) analysis identified 60 pg/mL as the optimal cutoff for detecting inflammation-mediated anemia (AUC: 0.76; 95% CI: 0.71–0.81; Supplementary Material 1), outperforming 50 pg/mL and 65 pg/mL. Therefore, IL-6 > 60 pg/mL was used to classify inflammation-mediated anemia and to estimate the fraction of anemia attributable to inflammation. In addition, for determining the overall prevalence of inflammation, we also applied the literature-based threshold of IL-6 > 50 pg/mL [22].

Iron deficiency anemia (IDA) was defined as Hb < 10.5 g/dL and ferritin below 12 ng/mL for those aged 6−23 months and <11 g/dL and ferritin below 12 ng/mL for those aged 24−59 months. For IDA without inflammation, the criterion of IDA and IL-6 ≤ 60 pg/mL was used. Inflammation without IDA was defined as IL-6 > 60 pg/mL with normal Hb. Inflammation with IDA was defined as IL-6 > 60 pg/mL with IDA.

## Analyses

For descriptive statistics, frequencies and percentages for categorical variables and mean ± standard deviation (SD) for continuous variables were used. For the evaluation of differences in Hb by WHO classifications, the two-sided t-test was applied. The chi-square test was used to compare anemia prevalence according to WHO guidelines. The proportion of anemia attributable to ID or inflammation was estimated as: $P_e (PR - 1)/ 1 + P_e (PR - 1)$. Where $P_e$ is the exposed proportion of the population (High ID or High IL-6) and RR was the relative risk adjusted for age, sex, and altitude [23]. We used the prevalence ratio (PR) from adjusted Poisson regression models to estimate RR. STATA 18.0 statistical package was used for data analysis. A p-value <0.05 indicated statistical difference.

**Table 1. Comparison of 2001 WHO Guidelines and 2024 Guidelines for the diagnosis of anemia among children aged <5 years.**

|  | 2001* WHO Guidelines | 2024** WHO Guidelines |
|---|---|---|
| Hemoglobin cutoff for anemia in children aged 6–59 months | <11 g/dL | – |
| Hemoglobin cutoff for anemia in children aged 6–23 months | – | <10.5 g/dL |
| Hemoglobin cutoff for anemia in children aged 24–59 months | – | <11 g/dL |
| **Anemia severity grading for children aged 6–59 months** |  |  |
| Mild | 10.9−10 g/dL | – |
| Moderate | 9.9−7 g/dL | – |
| Severe | <7 g/dL | – |
| **Anemia severity grading for children aged 6–23 months** |  |  |
| Mild | – | 9.5-10.4 g/dL |
| Moderate | – | 9.4-7.0 g/dL |
| Severe | – | <7 g/dL |
| **Anemia severity grading for children aged 24–59 months** |  |  |
| Mild | – | 10-10.9 g/dL |
| Moderate | – | 7-9.9 g/dL |
| Severe | – | <7 g/dL |
| **Hemoglobin adjustment factor for altitude of residence (in meters)** |  |  |
| 1-499 | 0 | 0 |
| 500-999 | 0 | 0.4 |
| 1,000-1,499 | 0.2 | 0.8 |
| 1,500−1,999 | 0.5 | 1.1 |
| 2,000-2,499 | 0.8 | 1.4 |
| 2,500−2,999 | 1.3 | 1.8 |
| 3,000-3,499 | 1.9 | 2.1 |
| 3,500−3,999 | 2.7 | 2.5 |
| 4,000-4,499 | 3.5 | 2.9 |
| 4,500−4,999 | 4.5 | 3.3 |

According to *old guidelines [3] or **updated WHO guidelines [2].

## Results

A total of 310 infants and children were included. There were 144 (46.5%) males, and 166 (53.5%) females, with 26.1% aged 6−23 months and 73.9% aged 24−59 months (Table 2). Age, altitude of residence, Hb concentration, updated adjusted Hb concentration, serum hepcidin, EPO, Ferritin, sTfR, and IL-6 were similar between both sexes (Supplementary Table 1).

### Prevalence of anemia by WHO Hb cutoffs

When Hb was not adjusted for altitude with any formula, the prevalence of anemia in infants and children aged 6–59 months was 5.2%, within the group, 87.5% of the cases were mild, and 12.5% moderate. This value was nearly eight-fold higher (42.2%) with Hb adjustment for altitude with the new WHO guidelines and was 50% using the old WHO guidelines for altitude adjustment (reduction of 8 percentage points, p = 0.053). According to the old anemia definition, 50% of children with anemia had mild anemia, 46.4% had moderate anemia, and 3.2% had severe anemia. Conversely, using the new WHO anemia definitions, the proportion of children with mild anemia was 59%, moderate anemia was 38%, and 3% of cases were severe anemia (Fig 1). Fig 2 shows the differences in anemia prevalence according to the old and new

**Table 2. Descriptive statistics of the sample of infants and children aged 6-59 months residing in the 13 districts of Puno, Peru.**

| Variables | N (%) |
|---|---|
| Age (months) | |
| 6-23 | 81 (26.1) |
| 24-59 | 229 (73.9) |
| Age (months)[&] | |
| Sex | 33.44 ± 14.2 |
| Male | 144 (46.5) |
| Female | 166 (53.5) |
| Hemoglobin* according to old WHO guideline [3][&] | 10.8 ± 0.1 |
| Hemoglobin* according to new WHO guideline [2][&] | 11.0 ± 0.1 |
| Biomarkers in serum[&] | |
| Ferritin (ng/mL) | 21.07 ± 1.5 |
| sTfR (mg/L) | 0.75 ± 0.03 |
| Erythropoietin (mIU/mL) | 22.3 ± 0.9 |
| Hepcidin (ng/mL) | 24.6 ± 1.1 |
| Interleukin-6 (pg/mL) | 44.5 ± 1.3 |
| Iron deficiency[#]<br>Inflammation (50pg/mL)[##] | 75 (24.2)<br>114 (36.8) |
| Inflammation (60 pg/mL)[###] | 67 (21.6) |

Some variables do not add up to 100% due to missing values.

[&]Quantitative variables (age, hemoglobin and biomarkers) expressed as mean ± SD. * Hemoglobin adjusted according to WHO 2001 or 2024 guidelines. [#]Iron deficiency: Serum ferritin <12 ng/ml. [##]Inflammation: IL-6 > 50 pg/mL. [###]Inflammation: IL-6 > 60 pg/mL Anemia was defined when Hb < 10.5 g/dl in infants aged 6–23 months and Hb < 11 g/dL for children aged 24–59 months. Hemoglobin was adjusted for altitude according to old guidelines [3] or updated WHO guidelines [2].

WHO guidelines, classified by age group, 6–23 months, the group with the greatest reduction after the cutoff points were updated, around 20 percentage points (p = 0.059), and the 24–59-month age group (p > 0.05).

With application of the updated WHO guidelines to define anemia in comparison to the old WHO guidelines, 131 children aged 6–59 months were classified as having anemia, 24 who had anemia by the old WHO guidelines were reclassified as having normal Hb values, and 155 maintained being categorized as having normal Hb values (Table 3).

## Hemoglobin concentrations by altitude adjustment

The unadjusted mean Hb concentration in children aged 6–59 months was 13.5 g/dL (SD ± 0.08). When applying the WHO correction factor of Hb for altitude of residence, the average Hb concentration according to the old WHO adjustment for altitude was 10.8 g/dL (SD ± 0.09) [3], and 11.03 g/dL (SD ± 0.08) using the new WHO criteria for children aged 6–59 months [2].

## Prevalence of ID and inflammation

The prevalence of ID and inflammation in children aged 6–59 months old was 24.2% and 21.6%, respectively (Table 2). The prevalence of ID was similar to the prevalence of inflammation among children aged 6–23 months (p = 0.653) and those aged 24–59 months (p = 0.999). ID was similar among children aged 6–23 months and those aged 24–59 months (p = 0.139). The prevalence of inflammation was also similar among those aged 6–23 months and those aged 24–59 months (p = 0.898).

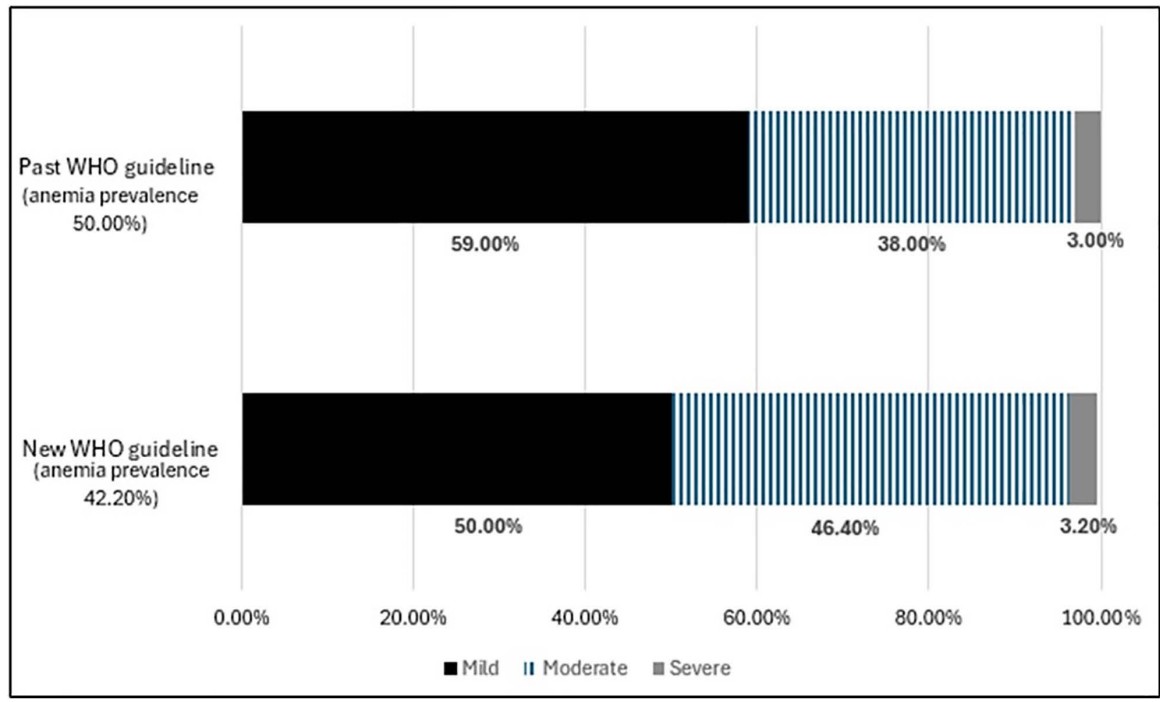

**Fig 1. Prevalences of anemia in infants aged 6 to 59 months residing in Puno, by category.** Black bar, mild anemia. Black lines bar, moderate anemia. And, gray bar, severe anemia calculated based on old and new WHO guidelines.

Supplementary Table 2 includes data in young children diagnosed as having normal Hb, mild, moderate, and severe anemia. For these children, 179 (57.7%) had normal Hb, 77 (24.8%) were classified as mild anemia, 50 (16.1%) as moderate anemia, and only 4 (1.4%) as severe anemia.

In these groups, serum hepcidin, EPO, SF, STfR and IL-6 were similar. The proportion of children with low SF was not different if they had normal Hb (0.21, 95% CI: 0.16–0.27), mild anemia (0.28, 95% CI 0.19–0.38), or moderate/severe anemia (0.37, 95% CI 0.25–0.49).

## Proportions of anemia etiologies

IDA was present in 13.2% children aged 6–59 months (19.7% among those aged 6–23 months and 10.9% among those aged 24–59 months). Among children aged 6–59 months, the prevalence of IDA without evidence of inflammation was 11.0% (16.1% among those aged 6–23 months and 9.2% for those aged 24–59 months [p > 0.05]) (Table 4).

Among children aged 6–59 months, the prevalence of inflammation-mediated anemia without evidence of ID was 19.4% (17.3% among children aged 6–23 months and 20.1% among children aged 24–59 months; Table 4). The attributable fraction of anemia associated with ID without inflammation was 27.5% for 6–59 months (34.2% among children aged 6–23 months and 24.7% for those aged 24–59 months).

The attributable fraction of inflammation-mediated anemia was 45.9%, while 26.6% of anemia was attributable to other non–iron-related causes (e.g., genetic, environmental factors) (Table 4).

There was also a non-significant trend for lower SF in more severe anemia (22.38 ± 1.93, 18.34 ± 1.11, 17.98 ± 1.57, 14.39 ± 3.0 for normal Hb, mild anemia, moderate anemia, and severe anemia respectively, p = 0.32). Similarly, no difference was observed between prevalence of inflammation between infants with normal Hb, mild, moderate and severe anemia (p = 0.22).

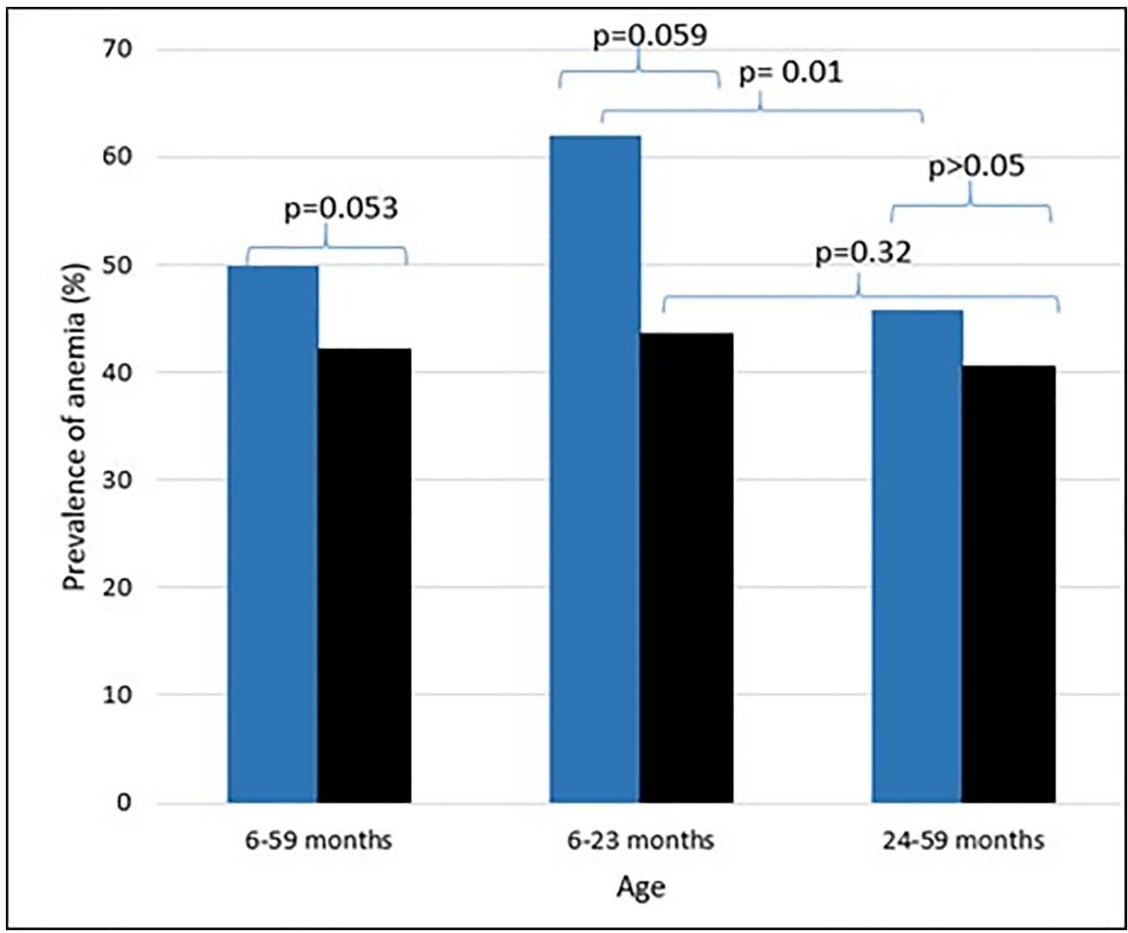

**Fig 2. Prevalence of anemia according to age groups 6 to 59, 6 to 23, and 24 to 59 months of age adjusted for old WHO guidelines (Anemia according to WHO 2001 guideline) and new WHO guidelines (Anemia according to WHO 2024 guideline) for both hemoglobin cutoff and altitude adjustment.** *All comparisons were made using the chi-square test.

**Table 3. Hemoglobin, iron, and inflammatory status in children aged 6-59 months using 2024 WHO anemia guidelines.**

| Biomarkers | Normal Hb, (n=155) | Anemic, (n=131) | Normal Hb after reclassification, (n=24) |
|---|---|---|---|
| Hb (g/dL)¥ | 12.2±0.1 | 9.7±0.1* | 10.9±0.1**,# |
| Hepcidin (ng/mL) | 23.9±1.7 | 25.3±1.51 | 25.1±1.9 |
| sTfR (mg/L) | 0.7±0.04 | 0.8±0.04 | 0.9±0.1 |
| Ferritin (ng/mL) | 23.5±2.7 | 18.6±1.05 | 19.6±2.9 |
| IL-6 (pg/mL) | 43.2±1.7 | 47.3±2.17 | 37.5±4.5# |
| EPO (mIU/mL) | 21.7±1.5 | 22.3±5.60 | 26.5±2.9 |

Data are reported as mean±SD. Hb: Hemoglobin; EPO: Erythropoietin; sTfR: Soluble transferrin receptor; IL-6: Interleukin-6. ¥Adjusted Hb (updated version) for altitude was calculated using the guidelines from WHO in 2024 [3]. *$p < 0.001$ between normal and anemic children. **$p < 0.001$ between normal and normal after reclassification children. #$p < 0.001$ between anemic and normal after reclassification children.

**Table 4. Prevalence of anemia, iron deficiency anemia (IDA), inflammation-mediated anemia and attributable fractions through unadjusted comparisons.**

|  | 6–59 Months (95% CI) | 6–23 Months (95% CI) | 24–59 Months (95% CI) |
|---|---|---|---|
| Anemia prevalence | 42.2 (36.7; 47.9) | 46.9 (35.7; 58.3) | 40.6 (34.2; 47.2) |
| IDA prevalence* | 13.2 (9.7, 17.5) | 19.7 (11.7; 30.0) | 10.9 (7.2; 15.6) |
| IDA w/o Inflammation* | 11.0 (7.7; 15.0) | 16.1 (8.8; 25.9) | 9.2 (5.7; 13.7) |
| Inflammation** w/o IDA | 19.4 (15.1; 24.2) | 17.3 (9.8; 27.3) | 20.1 (15.1; 25.9) |
| Inflammation** with IDA | 2.3 (0.9; 4.6) | 3.7 (0.8; 10.4) | 1.8 (0.5; 4.4) |
| **Attributable Fraction (AF)** | | | |
| AF ID w/o Inflammation | 27.5 (22.5; 32.7) | 34.2 (24.2; 45.8) | 24.7 (19.0; 31.0) |
| AF Inflammation w/o ID | 45.9 (40.2; 51.5) | 36.8 (25.4; 47.2) | 49.3 (42.7; 56.0) |
| Other AF## | 26.6 (21.8; 31.6) | 29.0 (19.6; 39.5) | 26.0 (20.5; 31.8) |

AF for ID was calculated as the prevalence of IDA without inflammation divided by the total anemia prevalence according to the previously described method [24],

#ID was defined when serum ferritin was < 12 ng/mL

*IDA was defined when serum ferritin was < 12 ng/mL and Hb below the WHO cutoff point and

**Inflammation when IL-6 was > 60 pg/mL.

## Other AF: unmeasured attributable factors.

ID was associated with anemia defined by adjustment of Hb for altitude [2]. Inflammation was not associated with anemia with adjustment of Hb for altitude (Table 5). Higher sTfR was associated with anemia with Hb adjusted for altitude. The proportion of anemia associated to ID was significant (p = 0.001) with a PR of 1.32 (95% CI 1.0–1.7). The proportion of anemia associated with inflammation was not different between children with mild and moderate/severe anemia (Table 6).

## Biomarkers of iron status in children according to inflammatory status

No differences were observed in serum hepcidin, EPO, sTfR, and ferritin between age groups categorized as having anemia with the 2001 WHO guidelines. The serum IL-6 values were lower in anemic children categorized as having normal Hb concentration with the 2024 WHO guidelines (Table 3). Serum hepcidin, sTfR, EPO, and ferritin were similar among

**Table 5. Unadjusted, bivariate analyses between the different candidate features and anemia as the outcome.**

| Variable | Model | | |
|---|---|---|---|
|  | Prevalence Ratio | p-value | 95% Confidence Interval |
| Age: 24–59 vs. 6–23 months | 0.9 | 0.313 | 0.7 - 1.1 |
| Sex: Males vs. Female | 1.0 | 0.791 | 0.8-1.3 |
| Hepcidin (ng/mL) | 1.0 | 0.559 | 0.9 - 1.0 |
| Epo (mIU/mL) | 1.0 | 0.992 | 0.9 - 1.0 |
| sTfR (mg/L) | 1.3 | 0.011 | 1.1 - 1.6 |
| ID: Yes vs. No | 1.4 | 0.008 | 1.1-1.8 |
| Inflammation: Yes vs. No | 1.1 | 0.633 | 0.8-1.5 |

Unadjusted, modified Poisson regression with robust variances. Outcome anemia according to WHO, 2024. *Iron deficiency: Serum ferritin <12 ng/mL. **Inflammation: IL-6 > 60 pg/mL.

**Table 6. Reduced Modified Poisson Regression models with proportion of anemia attributable to ID and inflammation for each level of severity of anemia according to WHO-2024.**

| Variable | Total anemia | | | Mild anemia | | | Moderate/Severe anemia | | |
|---|---|---|---|---|---|---|---|---|---|
| | PR | p-value | 95% CI | PR | p-value | 95% CI | PR | p-value | 95% CI |
| ID* | **1.32** | 0.022 | 1.0-1.7 | 1.35 | 0.143 | 0.9-2.0 | **1.93** | 0.005 | 1.2-3.0 |
| Inflammation** | 1.20 | 0.144 | 0.9-1.5 | 1.22 | 0.344 | 0.8-1.9 | 0.93 | 0.828 | 0.1-0.8 |

Anemia diagnosis and severity according to WHO guidelines, 2024. Model: Modified Poisson Regression with robust variances, included age and sex. PR: prevalence ratio. CI: confidence interval. *Iron deficiency: Serum ferritin <12 ng/mL. **Inflammation: IL-6 > 60 pg/mL.

children with normal Hb, those with anemia, and those with normal Hb after classification with the 2024 WHO guidelines (Table 3).

When infants were grouped according to serum IL-6 levels, those with IL-6 > 60 pg/mL had similar age, altitude of residence, Hb levels (adjusted or unadjusted), hepcidin, EPO, sTfR, and ferritin levels as those with IL-6 levels ≤60 pg/mL (Supplementary Table 3). No significant correlation was observed between serum IL-6 and SF ($R^2 = 0.0004$; p > 0.05), and no correlation was observed between serum IL-6 levels and serum hepcidin levels ($R^2 = 0.072$; p > 0.05).

## Discussion

This study assessed the prevalence of anemia and its prevalence or attributable factors in infants and young children living at high altitudes in Southern Peru, comparing WHO recommendations from 2001 and 2024 [2,3]. Application of the new WHO guidelines resulted in the reclassification of 15% of children aged 6–23 months as not having anemia who were previously classified as having anemia by the old WHO guidelines. Our findings highlight the impact of modifying Hb thresholds on anemia diagnosis among infants aged 6–23 months. The reduction in anemia prevalence aligns with recent studies from low-altitude populations, where the updated WHO cutoffs resulted in 20% less classification of cases of anemia [23].

Children reclassified as non-anemic under the updated WHO guidelines had iron status similar to those classified as anemic or with normal Hb levels. This suggests that Hb alone may not be an adequate marker for anemia diagnosis at high altitudes. A more reliable alternative could be arterial oxygen content, which includes Hb levels and oxygen saturation [24]. The WHO attributes 50% of anemia cases to ID, 42% to inflammation, and 8% to other factors, including micronutrient deficiencies, hemoglobinopathies, and genetic disorders [25].

However, in Southern Peru, one of the regions with the highest anemia prevalence, these proportions appear different.

Using the old WHO hemoglobin cutoff, 50.2% of anemia cases in our sample were attributed to non–iron deficiency causes [6]. In contrast, when applying the updated WHO hemoglobin threshold [2], 26.6% of anemia cases were attributed to non–iron deficiency causes in our study population. These values reflect changes in the number of children classified as anemic under different Hb thresholds, not differences in the underlying etiologic distribution of anemia causes.

In Puno, anemia cases attributed to other causes were common. No global reports indicate such high proportions of anemia due to unidentified causes. Further studies should explore additional micronutrients and genetic factors, such as hemoglobinopathies, to clarify these findings [26–28]. In our study, only 11% of anemia cases were due to ID overall. However, ID is an attributable fraction in 45.9% of the anemic population. This differs from prior studies [29,30] and exceeds previous estimates for Puno, where the proportion of anemia cases attributed to non-ID causes was already high under previous WHO criteria [5].

Given the multiple causes of anemia observed in Puno, iron supplementation alone may not be sufficient to reduce its burden. Understanding the relative contributions of different risk factors is essential for designing effective prevention

strategies. The fact that 87% of anemia cases in Puno were attributed to causes other than ID or inflammation raises critical questions that future research must address.

Although iron deficiency was present in 11% of the total population, among children with anemia, 27.5% of cases were attributable to ID, consistent with previous global studies, which reported that 25–30% of anemia cases are attributable to ID [30,31]. WHO estimates that 42% of children and 49% of non-pregnant women with anemia have ID-related anemia [31,32]. However, our results indicate that assuming 50% of anemia cases are due to ID may no longer be accurate. Iron supplementation and iron-fortified food programs have not substantially reduced anemia prevalence [33,34]. Recognizing this stagnation, WHO recommended in 2023 that anemia management strategies should be more comprehensive, addressing multiple contributing factors beyond iron deficiency [4].

Our findings suggest that the 2024 WHO altitude adjustment may require additional precision, as it may lead to misdiagnoses for anemia. The large proportion of anemia cases attributed to other causes may result from suboptimal Hb correction factors for altitude [6,26,27,34]. Additionally, diagnosing anemia based solely on Hb levels may be insufficient. A more comprehensive evaluation should include complete blood count parameters, such as mean corpuscular hemoglobin and volume, to assess ID, infection, or inflammation [35,36].

Some researchers propose adjusting Hb cutoffs for altitude using large national datasets that incorporate regional, ethnic, and generational factors [37,38]. Although the WHO acknowledges these approaches, the WHO maintains that further evidence is needed before revising its recommendations. Prior research across three continents (America, Asia, and Africa) has also questioned the necessity of adjusting Hb for altitude [26–28], and our study's findings support this movement to reconsider current recommendations for altitude adjustment.

Results from prior results in Peru suggest that the diagnosis of anemia (and anemia severity) may need to be based on biological and functional markers rather than fixed Hb cutoffs [38–40]. Studies show that in high-altitude populations, adjusting Hb for altitude affects serum ferritin's validity as a marker of iron reserves. The correlation between anemia and serum ferritin weakens when Hb is adjusted for altitude, reinforcing the need to reconsider this correction [41].

In addition to ID and inflammation, other potential contributors include deficiencies in micronutrients such as vitamin A and B12, as well as genetic conditions like thalassemia. A recent study in Puno found no significant differences in zinc, calcium, vitamin A, or folate intake among children with anemia, normal Hb levels, or elevated Hb levels [16]. However, anemic children had lower beta-carotene and ascorbic acid intake compared to non-anemic groups. This study initially identified only 4.7% of children as anemic, but after adjusting Hb for altitude, the reported anemia prevalence changed to 65.6% with reclassification [16]. These findings highlight the need for a more nuanced approach to anemia diagnosis and management, considering multiple risk factors beyond iron deficiency and reevaluating the appropriateness of altitude-based Hb adjustments.

The strengths of the present study include the enrollment of a population of the southern Peruvian region with a high prevalence of anemia, in which the 13 provinces of the Puno Region have been evaluated covering altitudes of 600–4,500 meters.

Moreover, we measured markers of iron status and a single inflammatory marker (IL-6), which allowed us to estimate the fractions of anemia attributable to iron deficiency and to inflammation.

The study has several limitations. First, we did not measure biomarkers that would have allowed the assessment of other micronutrient deficiencies, such as vitamin A, vitamin C, and zinc, nor markers that could help quantify the contribution of hereditary disorders and hemoglobinopathies to anemia in this region.

Furthermore, we cannot perform comparative analyses with studies that apply the BRINDA method (which uses CRP and AGP) because these biomarkers were not measured in our study.

A further limitation is the use of IL-6, rather than CRP and AGP, to assess inflammation, as no standardized IL-6 cutoff exists for defining inflammation-mediated anemia. This lack of harmonization limits the direct comparability of our results with studies based on CRP/AGP-defined inflammation.

However, the use of IL-6 also represents a methodological strength, as IL-6 is a key upstream regulator of hepcidin, the central hormone responsible for iron sequestration during inflammation. Because hepcidin-mediated iron restriction is the primary mechanism underlying inflammation-induced anemia, IL-6 provides a biologically relevant and mechanistically aligned marker of inflammatory activity. Moreover, IL-6 responds rapidly to acute inflammatory stimuli and is less influenced by chronic, low-grade inflammation compared with CRP or AGP, allowing more precise identification of acute inflammatory states in young children.

Because the study used convenience sampling rather than a population-based design, the attributable fraction estimates should be interpreted as internal to the study sample and not extrapolated to the general population. Nevertheless, the sample was drawn from a vulnerable population across all provinces of Puno, which enhances the representativeness of the findings within this specific context.

## Conclusions

The application of the update WHO guidelines to diagnose anemia resulted in a lower prevalence of anemia, especially in infants aged 6–23 months. However, the altitude correction factor that has been modified may result in a sizeable proportion of children with anemia without an etiology of anemia, though the assessment of additional etiologies may be required. The altitude correction factor for anemia may need reevaluation.

## Supporting information:

**Supplementary Material 1:**
(DOCX)

## Acknowledgments

We thank the families of Puno who allowed the development of the research project.

## Author contributions

**Conceptualization:** Cinthya Vásquez-Velásquez, Benita Maritza Choque-Quispe, Parminder S Suchdev, Chris A. Rees, Vilma Tapia, Yi-An Ko, Gustavo F. Gonzales.

**Data curation:** Cinthya Vásquez-Velásquez, Yi-An Ko.

**Formal analysis:** Cinthya Vásquez-Velásquez, Vilma Tapia, Yi-An Ko.

**Funding acquisition:** Benita Maritza Choque-Quispe.

**Investigation:** Benita Maritza Choque-Quispe.

**Methodology:** Vilma Tapia, Gustavo F. Gonzales.

**Software:** Yi-An Ko.

**Supervision:** Parminder S Suchdev, Chris A. Rees.

**Validation:** Yi-An Ko.

**Writing – original draft:** Cinthya Vásquez-Velásquez, Benita Maritza Choque-Quispe, Parminder S Suchdev, Chris A. Rees, Vilma Tapia, Yi-An Ko, Gustavo F. Gonzales.

**Writing – review & editing:** Cinthya Vásquez-Velásquez, Benita Maritza Choque-Quispe, Parminder S Suchdev, Chris A. Rees, Vilma Tapia, Yi-An Ko, Gustavo F. Gonzales.

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
