## [Decision Letter · Decision Letter 0]

28 Aug 2025

Dear Dr. Vásquez-Velásquez,

Thank you for submitting your manuscript to PLOS ONE. After careful consideration, we feel that it has merit but does not fully meet PLOS ONE’s publication criteria as it currently stands. Therefore, we invite you to submit a revised version of the manuscript that addresses the points raised during the review process.

We look forward to receiving your revised manuscript.

Kind regards,

Raquel Inocencio da Luz, Phd

Academic Editor

PLOS ONE

Journal Requirements:

“The present study was funded by the Vice-Rectorate of Research of the Universidad Nacional del Altiplano, Puno, Peru. Dr. Rees was supported by the National Institutes of Health (K23HL173694).”

“The present study was funded by the Vice-Rectorate of Research of the Universidad Nacional del Altiplano, Puno, Peru. Dr. Rees was supported by the National Institutes of Health (K23HL173694).”

“The authors declare no conflicts of interest.”

5. Thank you for uploading your study's underlying data set. Unfortunately, the repository you have noted in your Data Availability statement does not qualify as an acceptable data repository according to PLOS's standards.

Reviewers' comments:

Reviewer's Responses to Questions

**Comments to the Author**

1. Is the manuscript technically sound, and do the data support the conclusions?

Reviewer #1: Partly

Reviewer #2: Yes

2. Has the statistical analysis been performed appropriately and rigorously?

Reviewer #1: No

Reviewer #2: Yes

3. Have the authors made all data underlying the findings in their manuscript fully available?

Reviewer #1: Yes

Reviewer #2: Yes

4. Is the manuscript presented in an intelligible fashion and written in standard English?

Reviewer #1: No

Reviewer #2: Yes

Reviewer #1: The authors conducted a cross-sectional study of children <5 years old in the highlands of Peru to determine the effect of the updated 2024 WHO anemia guidelines on anemia prevalence and proportion of anemia attributable to iron deficiency, anemia, and other causes. Anemia was lower using the updated WHO guidelines; however, the authors question the methodology for altitude adjustment given their finding that most anemia is due to causes other than iron deficiency. Overall, this study provides interesting results and discussion on the updated WHO anemia guidelines. The organization and repetition of the manuscript could be improved to facilitate understanding and comparison of the results with other similar studies. I have outlined points for improvement below.

1. The organization of the methods and results should be carefully reviewed and revised. I found this paper very confusing to read. Many sentences seemed out of place and belonged under other section headers, or seemed to repeat information that was already stated. For example, lines 195-197 seem like they belong in the subsequent section. Lines 204-213 seem like they should go under “prevalence of anemia by WHO Hb cutoffs”. Lines 249-252 seem out of place in the section under ID and inflammation. There are several such instances.

2. Clarify the terminology used to describe Hb and anemia before and after using the new WHO guidelines. Wording such as in lines 227-228 “lower in the group initially anemic who has normal Hb levels after being anemic” makes very little sense. These children are not changing anemia status in time, but are classified differently using different criteria. Please clarify throughout so that it’s clear when you’re referring to classification under the new guidelines versus old guidelines. Furthermore, be clear about whether you’re referring to reclassification based on altitude or cut-off, as sometimes this wasn’t clear.

3. The discussion was repetitive. For example, lines 323-326, 342-348, and 366-370 say basically the same thing. It would be helpful to combine such statements into one concise paragraph.

4. There is almost no discussion of inflammation and its contribution to anemia in the discussion despite being a central focus of the results section. Please elaborate on how important inflammation is to anemia and potential causes of inflammation in children in this setting. Your abstract could be updated to reflect these changes.

Lines 42: Should say “due to”

Lines 89: Should say “of anemia among”

Lines 106-107: Add a justification for your sample size. Was a sample size calculation done?

Lines 108-113: Please rewrite this sentence for clarity.

Lines 128-133: Add section heading.

Lines 134-141: Suggest moving to the beginning of the methods.

Lines 166-167: Justify the use of IL-6 > 60 pg/mL. The reference does not appear to mention this cut-off and does not seem to be appropriate for justifying this cut-off value.

Lines 17-175: Did you adjust TBI for inflammation per BRINDA guidelines? See: 10.3945/ajcn.116.142307

Lines 227-228: This sentence is confusing. First, please reword per comment number 2 above. Second, based on your table, it looks like EPO was higher descriptively, but this was not statistically significant and is due to chance given your small n.

Line 231: Updated for altitude?

Line 236: Does Figure 1 provide updates based on both new cut-offs and altitude? Please clarify in text and figure legend.

Lines 266-271: Please clarify how these proportions are different from the proportions given in lines 260-265.

Lines 272-274: Seems repetitive with earlier results (lines 249-252). What is the difference?

Lines 342-345: Add the use of BRINDA adjustments to your methods. How did you adjust SF levels without the collection of CRP and AGP? Also, please justify why you did not use BRINDA for most of your estimates. This is the recommended methodology, but you seem to use it as a comparison. This should be made clearer.

Table 3: Define AF. Attributable fraction?

Table 3: I suggest separating the AF results from the others, either in another table or with headers, to clearly explain the difference between these prevalences.

Table 4: Are these all run as separate models?

Reviewer #2: Overall, the manuscript presents a relevant research subject, technically sound work, and the conclusions are supported by the data.

The main highlights of the paper are the importance to assess new guidelines, as they might foment changes in the prevalence numbers, and the value of data on population with specific conditions, such as cultural, socioeconomic and, as in this case, geographic uniqueness (high altitude).

This should be more emphasized in the conclusion.

Moreover, below are some suggestions for improvements.

Abstract

Lines 41-42

Improve writing to make it clearer that 6-23 and 24-59 months are brackets (subgroups) of the overall age interval (6-59 months). It is inferable, but it could be more objectively described.

Methods

Lines 120-126

The reference for CRED (15; Peruvian government site) indicates that the program is mandatory for all Peruvian children. If so, it would be better to report it. If not, does the children outside of the program not receive or are demanded a complete vaccination schedule? For non-Peruvians unfamiliar with the program, the description is a bit vague. It should either be a little bit clearer or just better summarized and referenced.

Lines 137-138

Beside the absolute number, consider adding the percentage of children in the Puno population, for perspective.

Line 139

The word “maintained” gives the impression that they are financially supported, but it doesn’t say by whom. Suggestion: “indigenous to”, “inhabiting”, or others.

Lines 157, 161, 173, 174

The sign before 4.5 g/dl, 3.3 g/dl and 4mg/kg ( - ) is a dash, a typo, minus/negative...?

Lines 154-164

An informative figure/table comparing the old and new criteria would be clearer and highly valuable for future consultation/citations.

Line 170

The acronym IA is not previously presented in the manuscript. Inflammatory anemia?

Results

Lines 195-197: it says the non altitude-adjusted anemia prevalence was 5.2%. Line 204 says it was 50%. It is confusing.

Lines 209-214

A bar graph would be invaluable to show that one of the main changes with the new guideline, more than just the total prevalence, is the classification distribution, such as the increase in mild and decrease in moderate anemia.

Line 298

“Relationship” might be a false cognate in this context. Correlation.

Discussion

Lines 303-305

“a greater reduction than achieved through years of universal iron supplementation”: as it is stated, it seems that updating the guidelines was a estrategy to reduce the anemia prevalence and colaborated with it, when, in fact, it just changed the threshold. It was a reclassification.

Table 1

“Quantitative variables expressed (age, hemoglobin and biomarkers) as mean ± SD”. Rewrite as “Quantitative variables (age, hemoglobin and biomarkers) expressed as mean ± SD”

Table 2

The line N should be inside the header line, such as: Normal Hb (n=155); Anemic (n=131); Normal Hb after reclassification (n=24).

“normal after update cutpoint children”: grammar is confusing. Maybe “normal after reclassification”?

**Do you want your identity to be public for this peer review?** For information about this choice, including consent withdrawal, please see our Privacy Policy

Reviewer #1: No

Reviewer #2: **Yes:** Douglas dos Santos Moreira

---

## [Author Response · Author response to Decision Letter 1]

4 Sep 2025

ANSWERS OF REVIEWERS

Manuscript ID: PONE-D-25-18153

Manuscript title: Changes in anemia and proportion of anemia associated with iron deficiency or inflammation in young children residing in Puno, Peru: Analysis using new World Health Organization guidelines for defining anemia

Dear Editor and Reviewers,

Thank you for the opportunity to revise and resubmit our article, “Changes in anemia and proportion of anemia associated with iron deficiency or inflammation in young children residing in Puno, Peru: Analysis using new World Health Organization guidelines for defining anemia” (PONE-D-25-18153). In our responses below, we have indicated where the corresponding changes were made in the revised manuscript. We are happy to address further questions that may arise.

We thank you very much for reviewing our revised submission. Please do not hesitate to contact us with any questions.

Journal Requirements

Author response: We appreciate the Editor’s input and guidance. We have ensured that our manuscript meets PLOS One’s style requirements as outlined in the link included here.

“The present study was funded by the Vice-Rectorate of Research of the Universidad Nacional del Altiplano, Puno, Peru. Dr. Rees was supported by the National Institutes of Health (K23HL173694).”

Author response: We have added the following to the Funding statement as requested, “There was no additional external funding received for this study.”

We have also added the amended Funding Statement in the cover letter.

“The present study was funded by the Vice-Rectorate of Research of the Universidad Nacional del Altiplano, Puno, Peru. Dr. Rees was supported by the National Institutes of Health (K23HL173694).”

Author response: We have added the following to the Funding Statement as requested, “The funders had no role in study design, data collection and analysis, decision to publish, or preparation of the manuscript.”

“The authors declare no conflicts of interest.”

Author response: We have updated the text as requested to now read, “The authors have declared that no competing interests exist.”

5. Thank you for uploading your study's underlying data set. Unfortunately, the repository you have noted in your Data Availability statement does not qualify as an acceptable data repository according to PLOS's standards.

Author response: We have uploaded the database to the ZENODO repository, which is listed among the Cross-disciplinary repositories permitted by PLOS One. In the manuscript, we have included both the DOI and the URL to ensure proper access and citation.

Author response: We have added the following to the Methods section and removed the other mention of the Ethical Approvals: “The study was conducted in accordance with the Declaration of Helsinki and approved by the Institutional Review Board (or Ethics Committee) of Universidad Peruana Cayetano Heredia (protocol code N°464-20-19).”

Author response: Thank you for pointing this out. There were no such requests.

Reviewer #1:

Observation 1: The organization of the methods and results should be carefully reviewed and revised. I found this paper very confusing to read. Many sentences seemed out of place and belonged under other section headers, or seemed to repeat information that was already stated. For example, lines 195-197 seem like they belong in the subsequent section. Lines 204-213 seem like they should go under “prevalence of anemia by WHO Hb cutoffs”. Lines 249-252 seem out of place in the section under ID and inflammation. There are several such instances.

Author response: We appreciate the Reviewer’s comment. In response, we have carefully revisited the Results and have rearranged text to provide clearer writing and a more cohesive presentation of our findings as marked in tracked changes.

Observation 2: Clarify the terminology used to describe Hb and anemia before and after using the new WHO guidelines. Wording such as in lines 227-228 “lower in the group initially anemic who has normal Hb levels after being anemic” makes very little sense. These children are not changing anemia status in time, but are classified differently using different criteria. Please clarify throughout so that it’s clear when you’re referring to classification under the new guidelines versus old guidelines. Furthermore, be clear about whether you’re referring to reclassification based on altitude or cut-off, as sometimes this wasn’t clear.

Author response: We have carefully reviewed the entire manuscript and have updated the language from “changes” to “reclassification” and have clarified if the change in classification was done based on cut-offs or altitude adjustment as recommended by the Reviewer.

Observation 3: The discussion was repetitive. For example, lines 323-326, 342-348, and 366-370 say basically the same thing. It would be helpful to combine such statements into one concise paragraph.

Author response: We thank the Reviewer for this comment. We have consolidated these three sentences into a new paragraph in the Discussion.

Observation 4: There is almost no discussion of inflammation and its contribution to anemia in the discussion despite being a central focus of the results section. Please elaborate on how important inflammation is to anemia and potential causes of inflammation in children in this setting. Your abstract could be updated to reflect these changes.

Author response: We appreciate the Reviewer’s comment. We have included the following in the Discussion in response to this comment, “The WHO attributes 50% of anemia cases to ID, 42% to inflammation, and 8% to other factors, including micronutrient deficiencies, hemoglobinopathies, and genetic disorders [28]. However, in Southern Peru—one of the regions with the highest anemia prevalence—these proportions appear different.”

“A more comprehensive evaluation should include complete blood count parameters, such as mean corpuscular hemoglobin and volume, to assess ID, infection, or inflammation [38, 39].”

“Moreover, markers of iron status were measured, as well as inflammatory markers, which allowed the calculation of the fraction of anemia attributable to ID and to inflammation.”

We have also included the following in the Abstract to emphasize our findings on inflammation: “The proportion of anemia due to ID was 27.5%, due to inflammation was 45.9%, …The new WHO guidelines did not substantially alter the estimated proportion of anemia associated with ID or inflammation.”

Observation 5: Lines 42: Should say “due to”

Author response: We have corrected this line in the Abstract as suggested. This line now reads, “The ratio of anemia due to iron deficiency (Ferritin <12 ng/mL) or inflammation (IL-6 >60 pg/mL) was estimated using adjusted Poisson regression models, reporting prevalence ratios (PR).”

Observation 6: Lines 89: Should say “of anemia among”

Author response: This has been corrected as requested. This line in the Introduction now reads, “Peru has one of the highest prevalences of childhood anemia throughout the Americas with historical estimates of anemia among young children aged 6 to 35 months as high as >70% in its highland regions and >40% across the country [7].”

Observation 7: Lines 106-107: Add a justification for your sample size. Was a sample size calculation done?

Author response: We have added the following to the Methods to describe our sample size. “Our sample size was determined by the convenience sample of 310 children that enrolled in our study. Post hoc, we determined that, at a significance level of 0.05 (two-sided) and a power of 80%, the minimum required sample size was 122 children per group (anemic and non-anemic) in order to detect a difference in the prevalence of anemia by the 2001 WHO guidelines and the 2024 WHO guidelines. The final recruited sample consisted of 310 children (179 non-anemic children and 131 anemic children).”

Observation 8: Lines 108-113: Please rewrite this sentence for clarity.

Author response: We have simplified this sentence in the Methods to read as follows, “We conducted our study in Puno, Peru since this region has one of the highest prevalences of anemia in the world (i.e., estimated 70% in children).”

Observation 9: Lines 128-133: Add section heading.

Author response: As suggested, we have added the following section heading “Consent and Ethical Approvals:”.

Observation 10: Lines 134-141: Suggest moving to the beginning of the methods.

Author response: We have moved the Study Setting section to follow the Study Design section as requested.

Observation 11: Lines 166-167: Justify the use of IL-6 > 60 pg/mL. The reference does not appear to mention this cut-off and does not seem to be appropriate for justifying this cut-off value.

Author response: Thank you for noting this. We have updated the reference to include a cutoff of IL-6 >60 pg/mL. We have also justified our use of a stricter cut-off for IL-6 in comparison to the referenced publication as follows in the Methods, “Prior studies have used an IL-6 value of >65 pg/mL to identify those with inflammation. However, as there is no reference standard for IL-6 levels in children, we applied a cutoff of >60 pg/mL in order to include more children with potential inflammation-mediated anemia.”

Observation 12: Lines 17-175: Did you adjust TBI for inflammation per BRINDA guidelines? See: 10.3945/ajcn.116.142307

Author response: Thank you for this comment and for including this reference. We used the same formula included in this publication. Additionally, thanks to the Reviewer, we have added this reference to the same section in the Methods.

Observation 13: Lines 227-228: This sentence is confusing. First, please reword per comment number 2 above. Second, based on your table, it looks like EPO was higher descriptively, but this was not statistically significant and is due to chance given your small n.

Author response: We thank the Reviewer for this comment. We have adjusted the wording in this sentence to now read, “Serum hepcidin, sTfR, EPO, and ferritin were similar among children with normal Hb, those with anemia, and those with normal Hb after classification with the 2024 WHO guidelines (Table 2).”

Observation 14: Line 231: Updated for altitude?

Author response: We thank the Reviewer for this comment. This line has been updated to read, “The prevalence of anemia with the 2011 WHO altitude adjustment correction was 50% in children aged 6-59 months, whereas with updated 2024 WHO recommended altitude adjustment correction the prevalence of anemia changed to 42%, resulting in a reduction of 8 percentage points (p=0.053).”

Observation 15: Line 236: Does Figure 1 provide updates based on both new cut-offs and altitude? Please clarify in text and figure legend.

Author response: We have added additional clarifying language in the legend for Figure 1 as suggested. This now reads, “Figure 1. Prevalence of anemia according to age groups 6 to 59, 6 to 23, and 24 to 59 months of age adjusted for 2001 WHO guidelines and 2024 WHO guidelines for both hemoglobin cutoff and altitude adjustment.”

Observation 16: Lines 266-271: Please clarify how these proportions are different from the proportions given in lines 260-265.

Author response: We appreciate the opportunity to clarify these sentences in the Results. In response to the Reviewer, we have changed the text to read, “Among children aged 6-59 months without evidence of inflammation, the prevalence of IDA was 11.6% (16.1% among those aged 6-23 months and 10.0% for those aged 24-59 months [p>0.05]) (Table 3).” And “Among children aged 6-59 months, the prevalence of anemia due to inflammation without evidence of ID was 19.4% (17.3% among children aged 6-23 months and 20% among children aged 24-59 months; Table 3).”

Observation 17: Lines 272-274: Seems repetitive with earlier results (lines 249-252). What is the difference?

Author response: We have removed these lines as they were redundant with previous results.

Observation 18: Lines 342-345: Add the use of BRINDA adjustments to your methods. How did you adjust SF levels without the collection of CRP and AGP? Also, please justify why you did not use BRINDA for most of your estimates. This is the recommended methodology, but you seem to use it as a comparison. This should be made clearer.

Author response: The statistical technique applied was the same as that used in BRINDA (i.e., ≥1 marker of iron and ≥1 inflammatory marker). However, we could not fully apply the BRINDA approach as CRP and AGP were not collected in our study. We have added the following to the Limitations in response to the Reviewer’s comment, “Although we used the approach outlined by BRINDA for TBI, we did not have data for CRP and AGP, so we were unable to adjust for inflammation according to this standard. However, we did use a proxy for adjustment of ≥1 marker of iron and ≥1 inflammatory marker.”

Observation 19: Table 3: Define AF. Attributable fraction?

Author response: We have added a subheading in Table 3 as requested that defines AF as follows, “Attributable Fraction (AF)”.

Observation 20: Table 3: I suggest separating the AF results from the others, either in another table or with headers, to clearly explain the difference between these prevalences.

Author response: We appreciate this recommendation and we have added a subheading to make this disti

---

## [Decision Letter · Decision Letter 1]

29 Oct 2025

Dear Dr. Vásquez-Velásquez,

Thank you for submitting your manuscript to PLOS ONE. After careful consideration, we feel that it has merit but does not fully meet PLOS ONE’s publication criteria as it currently stands. Therefore, we invite you to submit a revised version of the manuscript that addresses the points raised during the review process.

We look forward to receiving your revised manuscript.

Kind regards,

Raquel Inocencio da Luz, Phd

Academic Editor

PLOS ONE

Journal Requirements:

Reviewers' comments:

Reviewer's Responses to Questions

**Comments to the Author**

Reviewer #1: (No Response)

Reviewer #2: (No Response)

2. Is the manuscript technically sound, and do the data support the conclusions?

Reviewer #1: No

Reviewer #2: Yes

3. Has the statistical analysis been performed appropriately and rigorously?

Reviewer #1: No

Reviewer #2: Yes

4. Have the authors made all data underlying the findings in their manuscript fully available?

Reviewer #1: Yes

Reviewer #2: Yes

5. Is the manuscript presented in an intelligible fashion and written in standard English?

Reviewer #1: No

Reviewer #2: Yes

Reviewer #1: The authors addressed several of my prior comments; however, the organization within the text, as well as the definitions of iron-deficiency anemia, inflammation-mediated anemia, and other categorizations, are not well-defined and difficult to follow in the text. Additionally, the use of IL-6 for inflammation-mediated inflammation, which lacks a validated cut-off, is problematic.

1. While I appreciate the authors' efforts to improve organization, significant issues with clarity and structure remain. The results are misplaced within the subheadings, and there is still repetition.

a. Lines 184-186 and 204-208 report nearly identical unadjusted anemia prevalence (5.2% vs 5.1%). Are these referring to the same value? Please remove this unnecessary redundancy if so.

b. The section "Prevalence of Anemia by Altitude Hb Correction Factor" (lines 203-214) largely repeats information from the preceding section and should be consolidated or deleted.

c. Lines 249-253 (BRINDA method results) appear disconnected from the surrounding content and need better integration or explanation

2. Justification of the methods used for IL-6 and the cut-off chosen in lines 155-158. Your explanation to artificially adjust the cut-off to “include more children with potential inflammation-mediated anemia” is not scientifically justifiable.

a. It is not appropriate to arbitrarily create a cut-off value and falsely inflate the number of children with inflammation.

b. You cite two studies that use >65 pg/ml, but one of these studies doesn’t use this cut-off and the other appears to use a cut-off of 70 pg/ml. Please clarify. I found one paper that used a cut-off of 50 pg/ml: https://bmcnutr.biomedcentral.com/articles/10.1186/s40795-023-00748-3#Sec2.

c. An option is to use one of these cut-offs from the literature as the main analysis, based on which is most appropriate to your study population, and then conduct a sensitivity analysis at a different threshold and report how these results change your interpretation of the proportion of inflammation and inflammation-mediated anemia.

d. It’s essential to note the use of IL-6, rather than CRP and AGP, as a limitation of your study, since there is no established cut-off for IL-6 to determine inflammation-mediated anemia. This limits the comparability of your work to other studies.

3. In the methods, please clearly write how you define iron-deficiency anemia, inflammation-mediated anemia, and the use of the BRINDA approach. (lines 158-162 and 170-172)

a. I assume in your inflammation-mediated anemia definition, you meant to include an IL-6 cut-off. Add this. Also include whether this is regardless of ferritin level.

b. Keep consistent terminology. If you use inflammation-mediated anemia, then don’t change to inflammation-related anemia, as this just created confusion.

c. How did you use BRINDA if you didn’t measure CRP and AGP? Is this justifiable?

4. Inconsistency between prevalence of ID.

a. Lines 216 state ID prevalence is 24.2%, but lines 231-232 state it's 3.5%. Clarify why these are different.

5. Inconsistency in results using your cut-off of inflammation-mediated anemia versus the BRINDA method.

a. In lines 241-242, the “proportion of anemia associated with inflammation was 45.9%” and in lines 249-250 “the proportion of anemia from ID and inflammation according to the BRINDA method was 11% and 2.1%”. There is also a huge inconsistency with the proportion attributable to ID between your results and using the BRINDA approach (27.% vs 11%).

b. Explain the 2.5-fold difference in ID attribution (11% vs 27.5%) and the 22-fold difference in inflammation attribution (2.1% vs 45.9%) between methods.

c. Decide which of these will be the primary method and why. In the discussion, you state, “In our study, only 11% of anemia cases were attributable to ID, while 87% were linked to other causes.” (lines 298-299). This makes it seem like you are using the most dramatic of these two results to frame your discussion.

d. You then state in lines 307-309 that 27.5% of anemia is associated with ID, directly contradicting the paragraph before. Clarify.

6. Lines 270-271, unclear why you included an equation here. If describing correlation, using R2.

7. The sentence in lines 292-294 is confusing. As written, it sounds as if these proportions are from an original report from the WHO and then the updated WHO guidelines, but the updated guidelines did not do attribution of causes. If this 26.2% comes from your results, then be clear about that and do not add a citation to this number. Also, if you are specifying all causes beside ID, it should be 26.6%+45.9% based on your results.

8. In the strengths, you write that you measured inflammatory markers, as plural. But as I understand it you only measured one inflammatory marker, IL-6.

9. Lines 350-351, the used of the updated WHO guidelines is noted as a strength. This does not seem like a strength to me if the aim of the paper is to compare the old versus new guidelines.

10. Lines 357-358. I don’t think this is a valid rebuttal to this limitation. You did not use CRP and AGP, limiting the use of the BRINDA method, and the inflammatory marker you did use does not have a validated threshold. This must be included in the limitations.

11. Table 4 requires clarification:

a. The row "Anemia associated with ID" (31.3%) is undefined and doesn't correspond to any other reported value. Please clarify what this represents.

b. The attributable fractions don't sum to 100%. The text mentions 26.6% "other causes" but this isn't shown in the table.

c. The opposite age-pattern results between methods (BRINDA: younger children have more inflammation; your IL-6 method: older children have more inflammation) needs explanation.

Reviewer #2: The majority of recommendantion were addressed by the author. A few still lingers, as described below.

Table 2

“normal after update cutpoint children” was not corrected in the footnote. Change it to “normal after reclassification”

Figure 1

Although I do understand that it is a matter of choice from the authors, in my perception, severe/moderate/mild reclassification changes are better visualized in a piled bars chart. It will not include the changes in total prevalence though. Also, the percentage values (59%, 46,4%…) should be plotted (this recommendation applies for other charts too).

Figure 2

The legend “Anemia 2001/Anemia 2024” gives the impresssion that there were a temporal analysis, in different years, when it was just the year of the guidelines. Correct it for clarity.

**Do you want your identity to be public for this peer review?**  For information about this choice, including consent withdrawal, please see our Privacy Policy

Reviewer #1: No

Reviewer #2: **Yes:** Douglas dos Santos Moreira

---

## [Author Response · Author response to Decision Letter 2]

28 Nov 2025

Manuscript ID: PONE-D-25-18153R1

Tittle: Changes in anemia prevalence and the proportion of anemia associated with iron deficiency or inflammation in young children residing in Puno, Peru: Analysis using new World Health Organization guidelines for defining anemia

Dear Editor and Reviewers,

Thank you for the opportunity to revise and resubmit our article. In our responses below, we have indicated where the corresponding changes were made in the revised manuscript. We are happy to address further questions that may arise.

Reviewer #1: The authors addressed several of my prior comments; however, the organization within the text, as well as the definitions of iron-deficiency anemia, inflammation-mediated anemia, and other categorizations, are not well-defined and difficult to follow in the text. Additionally, the use of IL-6 for inflammation-mediated inflammation, which lacks a validated cut-off, is problematic.

ANSWER: We appreciate the Reviewer’s comments and the time they spent reviewing our manuscript again. We have addressed each of these comments in detail as outline below. We believe that the incorporation of their input has greatly strengthened the reporting of our work.

Observation 1. While I appreciate the authors' efforts to improve organization, significant issues with clarity and structure remain. The results are misplaced within the subheadings, and there is still repetition.

a. Lines 184-186 and 204-208 report nearly identical unadjusted anemia prevalence (5.2% vs 5.1%). Are these referring to the same value? Please remove this unnecessary redundancy if so.

ANSWER: These values were the same. In response to the Reviewer’s comment, we have removed the redundancy.

b. The section "Prevalence of Anemia by Altitude Hb Correction Factor" (lines 203-214) largely repeats information from the preceding section and should be consolidated or deleted.

ANSWER: The structure of the paragraphs has been modified and redundant content has been removed (i.e., this section was removed as recommended by the Reviewer).

c. Lines 249-253 (BRINDA method results) appear disconnected from the surrounding content and need better integration or explanation

ANSWER: All sections referring to the BRINDA method have been removed in response to other comments below.

Observation 2. Justification of the methods used for IL-6 and the cut-off chosen in lines 155-158. Your explanation to artificially adjust the cut-off to “include more children with potential inflammation-mediated anemia” is not scientifically justifiable.

a. It is not appropriate to arbitrarily create a cut-off value and falsely inflate the number of children with inflammation.

ANSWER: The criteria for using the selected cutoff point have been detailed in the manuscript in the Methods on page 7 as described here: “We used IL-6 to measure inflammation [21, 22]. As prior studies have used varying cutoffs of IL-6, our selected cutoff was based on the 75th percentile, which corresponds to the cutoff of IL-6>60 pg/mL. We assessed the optimal IL-6 level for the diagnosis of inflammation-mediated anemia, which demonstrated an area under the receiver operating characteristic curve of 0.79; 95% CI: 0.71-0.81 (Supplementary Material 1). Thus, an IL-6 level of >60 pg/mL was used for our analyses.”

IL-6 Inflammation anemia

AUC* 95%CI

50 pg/mL 0.59 0.50-0.68

60 pg/mL 0.76 0.71-0.81

65 pg/mL 0.73 0.68-0.79

*Area under the curve adjusted for sex and age.

Supplementary material 1. Area under the curve adjusted for sex and age for the diagnosis of inflammatory anemia, using different IL-6 cutoff points: A. 50 pg/mL, B. 60 pg/mL, and C. 65 pg/mL.

b. You cite two studies that use >65 pg/ml, but one of these studies doesn’t use this cut-off and the other appears to use a cut-off of 70 pg/ml. Please clarify. I found one paper that used a cut-off of 50 pg/ml: https://bmcnutr.biomedcentral.com/articles/10.1186/s40795-023-00748-3#Sec2.

ANSWER: We appreciate the heterogeneity in reporting of IL-6 cutoffs in prior studies. In response to the Reviewer’s input, we have performed additional analyses as above to determine the diagnostic capacity of IL-6 based on different cut-off points. In the previous response, we detailed the areas under the curve for the 50, 60, and 65 pg/mL points. We have also included this as part of the supplementary material.

c. An option is to use one of these cut-offs from the literature as the main analysis, based on which is most appropriate to your study population, and then conduct a sensitivity analysis at a different threshold and report how these results change your interpretation of the proportion of inflammation and inflammation-mediated anemia. Detailed in the previous response.

ANSWER: We have considered the calculation of inflammation prevalence using the cut-off points of 50 and 60 pg/mL reported in Table 2 (36.8% vs. 21.6%, respectively). We have also performed a comparative AUC analysis to determine which of the cut-off points is optimal for the diagnosis of inflammation-mediated anemia.

d. It’s essential to note the use of IL-6, rather than CRP and AGP, as a limitation of your study, since there is no established cut-off for IL-6 to determine inflammation-mediated anemia. This limits the comparability of your work to other studies.

ANSWER: Certainly, we have included in the discussion section, the use of Il-6 rather than CRP and AGP as a limitation that limit comparisons with other studies using these inflammatory markers.

“Furthermore, we cannot perform comparative analyses with studies that considered inflammatory biomarkers such as those applied by the BRINDA method (e.g., CRP and AGP), as these were not measured…”

Observation 3. In the methods, please clearly write how you define iron-deficiency anemia, inflammation-mediated anemia, and the use of the BRINDA approach. (lines 158-162 and 170-172)

a. I assume in your inflammation-mediated anemia definition, you meant to include an IL-6 cut-off. Add this. Also include whether this is regardless of ferritin level.

ANSWER: The following has been incorporated into the text: “Iron deficiency anemia (IDA) was defined as Hb <10.5 g/dL and ferritin below 12 ng/mL for those aged 6-23 months and <11 g/dL and ferritin below 12 ng/mL for those aged 24-59 months. For IDA without inflammation, the criterion of IDA and IL-6≤60 pg/mL was used. Inflammation without IDA was defined as IL-6 >60 pg/mL with normal Hb. Inflammation with IDA was defined as IL-6 >60 pg/mL with IDA.”

b. Keep consistent terminology. If you use inflammation-mediated anemia, then don’t change to inflammation-related anemia, as this just created confusion.

ANSWER: We appreciate the opportunity to provide added clarity. We have reviewed the manuscript and we now exclusively use “inflammation-mediated anemia”

c. How did you use BRINDA if you didn’t measure CRP and AGP? Is this justifiable?

ANSWER: Since BRINDA uses CRP and AGP, the BRINDA method analysis in our manuscript has been removed. Table 1 shows the hemoglobin cut-off points, and Table 2 and the methodology show the cut-off points for the iron and inflammation biomarkers used.

Observation 4. Inconsistency between prevalence of ID.

a. Lines 216 state ID prevalence is 24.2%, but lines 231-232 state it's 3.5%. Clarify why these are different.

ANSWER: Line 216 refers to general iron deficiency assessed in the entire sample (i.e., in the 311 children who had ferritin below the cutoff point [<12 ng/mL]). Within this population, there were both anemic (Hb below threshold) and non-anemic children. In order to minimize potential for confusion, lines 231-232 have been removed because the key results are those related to IDA.

Observation 5. Inconsistency in results using your cut-off of inflammation-mediated anemia versus the BRINDA method.

a. In lines 241-242, the “proportion of anemia associated with inflammation was 45.9%” and in lines 249-250 “the proportion of anemia from ID and inflammation according to the BRINDA method was 11% and 2.1%”. There is also a huge inconsistency with the proportion attributable to ID between your results and using the BRINDA approach (27.% vs 11%).

ANSWER: The analysis of the BRINDA method has been removed as above.

b. Explain the 2.5-fold difference in ID attribution (11% vs 27.5%) and the 22-fold difference in inflammation attribution (2.1% vs 45.9%) between methods.

ANSWER: These were the values obtained using the BRINDA method, which have been removed from the manuscript, leaving only the values calculated based on our measured markers. The attributable fractions would be 27.5% and IA 45.9%.

c. Decide which of these will be the primary method and why. In the discussion, you state, “In our study, only 11% of anemia cases were attributable to ID, while 87% were linked to other causes.” (lines 298-299). This makes it seem like you are using the most dramatic of these two results to frame your discussion.

ANSWER: In response to the Reviewer, it has been specified that the prevalence of ID was 11% in the total population as follows: “In our study, only 11% of anemia cases were due to ID overall. However, ID as an attributable factor in 45.9% of the anemic population...”

d. You then state in lines 307-309 that 27.5% of anemia is associated with ID, directly contradicting the paragraph before. Clarify.

ANSWER: The value of 27.5% corresponds to the attributable factor, as specified in the text, which differs from the prevalence since the prevalence considers the entire child population evaluated and the attributable factor only considers children diagnosed with anemia.

Observation 6. Lines 270-271, unclear why you included an equation here. If describing correlation, using R2.

Answer: Thanks for the comment. We have removed the equation and only R2 is now displayed.

Observation 7. The sentence in lines 292-294 is confusing. As written, it sounds as if these proportions are from an original report from the WHO and then the updated WHO guidelines, but the updated guidelines did not do attribution of causes. If this 26.2% comes from your results, then be clear about that and do not add a citation to this number. Also, if you are specifying all causes beside ID, it should be 26.6%+45.9% based on your results.

Answer: Thank you for your comment. What we wanted to point out is that although we report attributable fractions without considering ID, the group of anemic patients used in the calculation differs due to the update of the Hb cutoff point. We have changed the following part of the paragraph for better understanding:

“50.2% of anemia cases were attributed to non-ID causes using the old WHO Hb cutoff. [6] Conversely, 26.6% of anemia cases were attributed to non-ID causes when the current Hb cutoff was used. [2]”

Observation 8. In the strengths, you write that you measured inflammatory markers, as plural. But as I understand it you only measured one inflammatory marker, IL-6.

Answer: Thank you for pointing that out, it has been changed.

Observation 9. Lines 350-351, the used of the updated WHO guidelines is noted as a strength. This does not seem like a strength to me if the aim of the paper is to compare the old versus new guidelines.

Answer: Thank you for pointing that out. This text has been removed as a result of this recommendation.

Observation 10. Lines 357-358. I don’t think this is a valid rebuttal to this limitation. You did not use CRP and AGP, limiting the use of the BRINDA method, and the inflammatory marker you did use does not have a validated threshold. This must be included in the limitations.

Answer: All mention of the BRINDA method has been removed from our manuscript in response to the Reviewer’s comments.

Observation 11. Table 4 requires clarification:

a. The row "Anemia associated with ID" (31.3%) is undefined and doesn't correspond to any other reported value. Please clarify what this represents.

ANSWER: The row on anemia associated with ID has been removed, as this is included in the total anemia prevalence.

b. The attributable fractions don't sum to 100%. The text mentions 26.6% "other causes" but this isn't shown in the table.

ANSWER: The specific percentage of the other factors has been calculated and included in Table 4, with a total sum of 100% for each age group. It has also been specified that genetic alterations, deficiencies of other vitamins, and alterations due to environmental pollution were not specifically measured. They were only mentioned in general terms in the text.

c. The opposite age-pattern results between methods (BRINDA: younger children have more inflammation; your IL-6 method: older children have more inflammation) needs explanation.

ANSWER: The section of the text that mentioned the BRINDA method has been removed.

Reviewer #2: The majority of recommendation were addressed by the author. A few still lingers, as described below.

Observation 12. Table 2 “normal after update cutpoint children” was not corrected in the footnote. Change it to “normal after reclassification”

Answer: Thank you for pointing that out. This has been changed as recommended.

Observation 13. Figure 1 Although I do understand that it is a matter of choice from the authors, in my perception, severe/moderate/mild reclassification changes are better visualized in a piled bars chart. It will not include the changes in total prevalence though. Also, the percentage values (59%, 46,4%…) should be plotted (this recommendation applies for other charts too).

Answer:

Figure 1. Prevalences of anemia in infants aged 6 to 59 months residing in Puno, by category. Black bar, mild anemia. Black lines bar, moderate anemia. And, gray bar, severe anemia calculated based on old and new WHO guidelines.

We have kept Figure 2 because the arrangement of the bars allows us to make statistical comparisons more easily. Figure 3 was removed because it contains information described in Table 4.

Observation 14. Figure 2 The legend “Anemia 2001/Anemia 2024” gives the impression that there were a temporal analysis, in different years, when it was just the year of the guidelines. Correct it for clarity.

Answer: Thank you for pointing that out. In response, it has been changed accordingly.

---

## [Editor Report · Decision Letter 2]

1 Dec 2025

Dear Dr.  Vásquez-Velásquez,

Thank you for submitting your manuscript to PLOS ONE. After careful consideration, we feel that it has merit but does not fully meet PLOS ONE’s publication criteria as it currently stands. Therefore, we invite you to submit a revised version of the manuscript that addresses the points raised during the review process.

We look forward to receiving your revised manuscript.

Kind regards,

Raquel Inocencio da Luz, Phd

Academic Editor

PLOS ONE

Journal Requirements:

Additional Editor Comments:

Even though progress has been made to your manuscript, there are still substantial comments from the reviewers. Carefully revise step by step, responding carefully to the inquiries of the reviewers. There are still some scientific flaws that need improvement as stated by the reviewers.

---

## [Author Response · Author response to Decision Letter 3]

10 Dec 2025

Thank you very much for your review. The attached document describes each of the points that have been noted.

---

## [Decision Letter · Decision Letter 3]

20 Jan 2026

Changes in anemia prevalence and the proportion of anemia associated with iron deficiency or inflammation in young children residing in Puno, Peru: Analysis using new World Health Organization guidelines for defining anemia

PONE-D-25-18153R3

Dear Authors,

We’re pleased to inform you that your manuscript has been judged scientifically suitable for publication and will be formally accepted for publication once it meets all outstanding technical requirements.

Kind regards,

Raquel Inocencio da Luz, Phd

Academic Editor

PLOS One

Additional Editor Comments (optional):

Reviewers' comments:

Reviewer's Responses to Questions

**Comments to the Author**

Reviewer #1: All comments have been addressed

2. Is the manuscript technically sound, and do the data support the conclusions?

Reviewer #1: Yes

3. Has the statistical analysis been performed appropriately and rigorously?

Reviewer #1: Yes

4. Have the authors made all data underlying the findings in their manuscript fully available?

Reviewer #1: Yes

5. Is the manuscript presented in an intelligible fashion and written in standard English?

Reviewer #1: Yes

Reviewer #1: (No Response)

**Do you want your identity to be public for this peer review?** For information about this choice, including consent withdrawal, please see our Privacy Policy

Reviewer #1: **Yes:** Nathalie Lambrecht

---

## [Editor Report · Acceptance letter]

PONE-D-25-18153R3

PLOS One

Dear Dr. Vásquez-Velásquez,

I'm pleased to inform you that your manuscript has been deemed suitable for publication in PLOS One. Congratulations! Your manuscript is now being handed over to our production team.

Kind regards,

on behalf of

Dr Raquel Inocencio da Luz

Academic Editor

PLOS One